# ACTAS: A New Framework for Mechanical and Frictional Characterization in Axisymmetric Compression Test

**DOI:** 10.3390/ma16010441

**Published:** 2023-01-03

**Authors:** Soheil Solhjoo

**Affiliations:** ENTEG, Faculty of Science and Engineering, University of Groningen, Nijenborgh 4, 9747 AG Groningen, The Netherlands; soheil@solhjoo.com

**Keywords:** axisymmetric compression test, flow stress curve, friction factor, foldover, barreling, meshfree method

## Abstract

There are two common methods to interpret the results of an Axisymmetric Compression Test (ACT): the Cylindrical Profile Model (CPM) and the Avitzur model; however, both of the two and all other models available in the literature ignore the unavoidable foldover phenomenon, which breaks the models to provide reliable friction-free flow stress curves. Here, a novel numerical framework (called ACTAS) is presented that incorporates the foldover. ACTAS can be used to both simulate and analyze ACT. Ten finite element models are used to benchmark ACTAS. The results show the reliability of the proposed method in estimating the average and pointwise stress-strain curves and friction factors. Moreover, a new solution is provided by coupling the CPM and the Avitzur model (called A-CPM), to obtain reliable average flow curves even after the onset of foldover.

## 1. Introduction

Axisymmetric Compression Test (ACT) is a crucial physical simulation to characterize material’s mechanical, microstructural, and frictional behaviors. The test provides “raw” measurements, including the shape of the deformed sample and the load-displacement curve, which require post-processing to be converted into “computed” stress, strain, strain rate, and friction. Currently, the available post-processing methods are mostly simplified closed-form solutions of ACT, which have narrow applications due to their limiting assumptions. Therefore, the results are no longer reliable if the methods are applied to tests beyond their limits. The readers are referred to a recent work of Khoddam, Solhjoo, and Hodgson [1] for a thorough review on the topic.

The most common way to convert load-displacement curves into “friction-free” stress-strain curves is to assume a uniform deformation within the sample by employing a Cylindrical Profile Model (CPM); see, for example, [2]. The CPM disregards barreling and shear deformation in the sample, among other simplifying assumptions. Moreover, although friction is incorporated into the CPM, it does not provide any methods to evaluate friction. Therefore, to use the CPM in the analysis of ACT, the friction must be negligible. This situation can be provided by either altering the test design [3] (see, e.g., Rastegaev [4], Nadai [5], and Chen et al. [6]) or proper lubrication, e.g., using polymers and biological tissues [7]. However, none of these adjustments eliminate friction to test metals with any significant large deformation relevant to metal forming processes. In a recent publication, Khoddam et al. [8] compared the CPM with a series of Finite Element (FE) models for a material with a monotonic flow curve for six different samples. They found that CPM underestimates the maximum effective stress, strain and strain rate. Additionally, when looking for a representative tracing point to validate the theories, they found that the CPM is incapable of fully identifying the flow curves at any sample point. More importantly, they reported that the CPM could lead to misleading results, such as the false identification of dynamic recovery and dynamic recrystallization, despite the monotonic nature of their reference samples.

Another common method for analyzing ACT is Avitzur’s limit analysis [9]. To use this model, its barreling parameter $b_{A}$ must be approximated first. A solution for $b_{A}$ was first proposed by Ebrahimi and Najafizadeh [10] and later by Solhjoo [11]. The value of $b_{A}$ can be approximated with numerous assumptions; see, e.g., the kinematic analysis of the Avitzur model, resulting in fifteen solutions for $b_{A}$ [12]. In another attempt, Yoa et al. [13] used numerical data and defined $b_{A}$ as a function of the geometry of the initial sample, the friction factor, and the mechanical behavior of the material. Despite the wide use of the Avitzur model, the validation attempts showed its unreliability in estimating the friction factor [11,14,15]. The possible reasons for the deficiency of the Avitzur model are discussed to be the unrealistic geometry of the deformed sample and, more importantly, the model’s inability to resolve the unavoidable side-surface foldover phenomenon [12,14]. The model’s reliability in estimating flow curve from ACT has yet to be assessed, which is covered in this work.

Additionally, there are a few less common methods to study ACT. Recently, Vuppala et al. [16] suggested an iterative approach. In this method, FE models are iteratively solved at any given strain. The flow stress is adjusted to match the deformation load of the simulation to that of the experiment. In this approach, friction is assumed to be known. Some other methods have been specifically developed to study friction. For example, Khoddam et al. [17] proposed an incremental approach based on an exponential profile model [18] to estimate friction factor; however, due to the limitations of the employed velocity field and the applied method, it is unable to capture foldover.

Foldover is inevitable in ACT for simulating any industry-relevant plastic deformation [12]. This complex phenomenon can be compared to necking in a tensile test and requires a specially developed theory to capture it in the model. It is crucial to properly post-process the test data after the onset of foldover [1,12]. For estimating the friction in the presence of foldover, Yang et al. [19] proposed a computationally demanding Finite Element Method (FEM-)based inverse solution using a multi-population genetic algorithm. This method is developed for studying viscoelastic materials. However, there is a less expensive method to study the test of plastically deformed metals: one can employ the closed-form solution of Ettouney and Stelson to estimate the foldover [20] and subsequently estimate the friction conditions using an FEM-based calibration method [21].

Avitzur and Kohser tried a two-zone velocity field [22,23] to account for the foldover in an attempt. Their new method simulates ACT, for which more accurate and versatile ways are available to incorporate barreling and foldover. So far, researchers use FE models to produce the most realistic results [24]. Yet, some flow-based models offer comparable solutions, e.g., the four-zone samples of Dadras and Thomas [25] or of Hou et al. [26]. All these methods are direct solutions to ACT, which means that they require the mechanical behavior of the material and the tool-sample interfacial friction to simulate ACT.

The current investigation aims to build a framework for a virtual laboratory based on the upper bound method to analyze ACT. It includes an ACT simulator (ACTS) and an analyzer (ACTA). The analyzer is the unique feature of this framework, which takes the raw measurements of the test as input and performs a reliable characterization of the mechanical behavior of the sample and the friction between the sample and the tool. The simulator, like others, requires the stress-strain curve and friction factor to predict the deformation load, the deformed shape of the sample, and the distribution of the state variables (e.g., stress, strain, and strain rate) throughout the sample.

## 2. Problem Overview and the Current Common Solutions

Figure 1 illustrates a schematic of ACT’s sample geometry and its key parameters used in ACT theories. This 2D illustration denotes a zero angular velocity, a common assumption of all current ACT theories. The figure also depicts a parameter *F* corresponding to the unavoidable foldover phenomenon, ignored in most theories. It is illustrated due to its coverage in the proposed method of the present study.

As mentioned in the Introduction, the two most used theories for interpreting ACT are the CPM and the one proposed by Avtizur. In the rest of this section, these two theories are briefly overviewed. Additionally, a possible connection between the two is suggested.

### 2.1. Cylindrical Profile Model

The CPM suggests a uniform distribution of state variables throughout the sample and neglects both barreling and foldover. Using different methods, such as slab analysis or upper bound for axisymmetric and homogeneous upsetting (see, e.g., [27]), the ratio of the average deformation pressure $p_{\mathrm{ave}}$ to the average stress $\bar{\sigma }$ can be found to be:(1)$$\frac{p_{\mathrm{ave}}}{\bar{\sigma }}=1+\frac{2}{3\sqrt{3}}\frac{m\bar{R}}{H},$$
with $\bar{R}=R_{0}\sqrt{H_{0}/H}$ being the effective radius. In this model, *m* is friction factor at constant interfacial shear stress with limits of 0≤m≤1. The friction factor *m* must be known beforehand, while the model does not provide any tool for estimating *m*. These limitations render the CPM useful only when the friction is negligible and the sample does not show barreling.

### 2.2. Avitzur Model

The Avitzur model is based on the upper bound solution for the average deformation pressure. It has a velocity field in its core, which introduces an arbitrary coefficient of “barreling parameter” $b_{A}$ to represent the barreling of the deformed sample. The theory can be used if $b_{A}\gg b_{A}^{n}$ for n≥2, such that $b_{A}^{n}\approx 0$ and $b_{A}<2\sqrt{3}H/\bar{R}$ [9,12]. This model connects the deformation pressure to the average stress by:(2)$$\frac{p_{\mathrm{ave}}}{\bar{\sigma }}=\frac{8}{g}\left({\left(\frac{1}{12}+g^{2}\right)}^{3/2}-g^{3}-m{\left(24\sqrt{3}\left(1-e^{b_{A}/2}\right)\right)}^{-1}\right),$$
with $g=H/(b_{A}\bar{R})$. Furthermore, the friction factor can be defined as a function of the barreling parameter:(3)$$m=\frac{3\sqrt{3}}{2}\frac{\bar{R}}{H}{\left(\frac{6}{b_{A}}-1\right)}^{-1}.$$ Therefore, if the barreling parameter can be obtained through measurements, the model can be used to interpret the ACT results. There are numerous ways to approximate $b_{A}$: see, e.g., [10,11,12,13]. However, regardless of the approximation method, this model provides highly unreliable estimations of the friction factor [11,12,28]. As mentioned earlier, the accuracy of its estimated flow curves has yet to be assessed.

### 2.3. The Connection between Cylindrical Profile and Avitzur Models

It is easy to show the differences between the geometry and the distribution of the state variables in the deformed samples of the CPM and the Avitzur model; see, e.g., [14]. Yet, in connecting $p_{\mathrm{ave}}$ and $\bar{\sigma }$, both of these models take a similar route to derive their formulations by ignoring the barreled geometry. In other words, the Avitzur model is essentially a CPM due to its assumption of a constant $\bar{R}$ as the upper limit of integration in calculating the internal deformation power and friction losses [9]. This assumption behind the derivation of Equation (Equation 2) suggests a possibility of connecting the two models, for example, by inserting the friction factor of the Avitzur model (Equation (Equation 3)) into the CPM (Equation (Equation 1)), resulting in:(4)$$\frac{p_{\mathrm{ave}}}{\bar{\sigma }}\approx 1+{\left(\frac{\bar{R}}{H}\right)}^{2}{\left(\frac{6}{b_{A}}-1\right)}^{-1}.$$

While the analytical comparison between Equations (Equation 2) and (Equation 4) is not trivial by any means, numerical comparisons for various values of the aspect ratio $\bar{R}/H$ and the barreling parameter help us to make an assessment. Figure 2 shows the ratio of Equation (Equation 4) to Equation (Equation 2) for the values of $b_{A}\in \left[0,1\right]$ and $\bar{R}/H\in \left[0.1,1\right]$, which safely range way beyond any practical limits [12]. The ratio varies between 1 and 1.02, suggesting a close correlation between the two models even under significant frictional conditions. Therefore, one can use the CPM approximated by the Avitzur model to analyze ACT results without a pre-known friction factor. Hereafter, I call it A-CPM (i.e., Equation (Equation 4)). In this paper, A-CPM is evaluated for various samples. This assessment reflects the reliability of both the CPM and the Avitzur model in their original forms.

## 3. Solving the Nonuniformity Problem

Several assumptions are claimed to be the sources of uncertainty of current ACT theories and their closed-form solutions. For example, (1) they assume a uniform constitutive behavior throughout the sample, and (2) their uniform description of material flow makes them intrinsically unable to describe the sample’s flow after the onset of unavoidable foldover. There are also a few successful studies on simulating ACT using flow-based models that incorporated the foldover phenomenon [25,26]. These models are carried out using numerical methods of dynamically evolving geometries.

ACTAS is a framework that uses a meshfree-based numerical method to solve both direct and inverse problems of ACT with an incremental approach. The selected form of the numerical method provides the potential for:dividing the sample into any desired number of zones,conditionally moving the zones’ boundaries,simulating the nonlinear material behavior, andsimulating the discontinuities and singularities occurring after the onset of the foldover by handling the creation, distortion, and destruction of nodes.

In the next section, the direct solution (ACTS) is first built to identify the tool’s limitations. Like other direct solutions, ACTS requires the input of flow curve and friction factor. Considering that other well-developed methods, such as FEM, can rigorously simulate ACT, one may question the reason for developing a new one. The primary reason is to build a foundation for the inverse solution ACTA. For a working ACTS, ACTA must be able to perform an accurate analysis; only then, performing ACTA on other data sets can be justified.

## 4. Mathematical Implementation of ACTAS

Investigating ACT with ACTAS requires three stages: preprocessing, processing, and postprocessing. The preprocessing stage is to generate the geometric model and to establish the problem domain and boundaries using scattered nodes. Contrary to the FEM, which requires a mesh for defining the information of the connected nodes, the nodes of the meshfree method, called field nodes, do not require any information on such a relationship for estimating the unknown functions of field variables. Moreover, the initial values of the state variables and the constitutive behavior of the material should be established at this stage. Once the problem is set, the solver (ACTS or ACTA) performs the calculations within an incremental procedure, preparing the results for postprocessing.

As mentioned in Section 3, both ACTAS modules use the Upper Bound Theorem (UBT) to solve the deformation load of ACT for an assigned velocity field. This section describes the UBT analysis of ACT first and then the mathematical implementations of both ACTS and ACTA.

### 4.1. Total Power Dissipation and Deformation Load

The upper bound theorem states that, among all kinematically admissible velocity fields, the actual one minimizes the total power dissipation due to plastic deformation $\dot{E}$. For a system where there is no predetermined body traction, such as the ACT with free side surfaces, $\dot{E}$ can be expressed as:(5)$$\dot{E}={\dot{E}}_{d}+{\dot{E}}_{f},$$
where ${\dot{E}}_{d}$ and ${\dot{E}}_{f}$ are the internal power of plastic deformation and the frictional power at the boundaries of velocity discontinuity (e.g., tool-workpiece interfaces), respectively. In a general case of a multizone velocity field, these powers are written as:
(6a)$$\begin{array}{cc} {\dot{E}}_{d} & =\sum_{i}\int_{V_{i}}^{}\sigma \dot{\bar{\varepsilon }}\;dV_{i}\;\mathrm{and} \end{array}$$
(6b)$$\begin{array}{cc} {\dot{E}}_{f} & =\sum_{i}\int_{S_{i}}^{}\tau \left|\Delta v_{S_{i}}\right|\;dS_{i}, \end{array}$$
where the index *i* indicates the zone *i*, $\left|\Delta v_{S_{i}}\right|$ is the absolute velocity discontinuity at the boundary of zone *i* with respect to its neighboring zones, $\dot{\bar{\varepsilon }}$ is nodal effective strain rate, σ is nodal stress, and τ is nodal shear stress at the boundaries of zone *i*.

For the frictional power, which is meaningful for the shearing interfaces, the shear stress would be described by incorporating a functional form for friction, commonly as τ=mκ at the tool-workpiece interface and τ=κ (i.e., m=1) for internal surfaces of velocity discontinuity, where κ is material’s shear flow stress. For the case of a one-zone velocity field, these powers can be simplified to:
(7a)$$\begin{array}{cc} {\dot{E}}_{d} & =2\pi \int_{0}^{H/2}\int_{0}^{R(z)}\sigma \dot{\bar{\varepsilon }}rdrdz\;\mathrm{and} \end{array}$$
(7b)$$\begin{array}{cc} {\dot{E}}_{f} & =4\pi \int_{0}^{R_{T}}\tau U_{r}\left(r,H/2\right)rdr, \end{array}$$
with R(z) being the radial profile of the barreled sample at any given height *z*.

The calculation of ${\dot{E}}_{d}$ requires the effective strain rate, which could be obtained from the strain rate components:(8)$$\dot{\bar{\varepsilon }}=\sqrt{2/3\left({\dot{\varepsilon }}_{rr}^{2}+{\dot{\varepsilon }}_{\varphi \varphi }^{2}+{\dot{\varepsilon }}_{zz}^{2}+1/2\left({\dot{\varepsilon }}_{rz}^{2}+{\dot{\varepsilon }}_{r\varphi }^{2}+{\dot{\varepsilon }}_{z\varphi }^{2}\right)\right)}.$$ These strain rate components can be derived from the velocity components using the following relations:
(9a)$$\begin{array}{cccc} & {\dot{\varepsilon }}_{rr} & & =\partial U_{r}/\partial r, \end{array}$$
(9b)$$\begin{array}{cccc} & {\dot{\varepsilon }}_{\varphi \varphi } & & =\frac{1}{r}\left(U_{r}+\partial U_{\varphi }/\partial \varphi \right), \end{array}$$
(9c)$$\begin{array}{cccc} & {\dot{\varepsilon }}_{zz} & & =\partial U_{z}/\partial z, \end{array}$$
(9d)$$\begin{array}{cccc} & {\dot{\varepsilon }}_{rz} & & =\partial U_{r}/\partial z+\partial U_{z}/\partial r, \end{array}$$
(9e)$$\begin{array}{cccc} & {\dot{\varepsilon }}_{r\varphi } & & =\frac{1}{r}\left(\partial U_{r}/\partial \varphi -U_{\varphi }\right)+\partial U_{\varphi }/\partial r, \end{array}$$
(9f)$$\begin{array}{cccc} & {\dot{\varepsilon }}_{z\varphi } & & =\frac{1}{r}\partial U_{z}/\partial \varphi +\partial U_{\varphi }/\partial z. \end{array}$$ The strain rate components ${\dot{\varepsilon }}_{ij}={\dot{\varepsilon }}_{ji}$ for any *i* and *j* of directions {r,z,φ}.

The power dissipation equation may contain a set of various unknowns $\hat{x}$ whose values, in any given state of the problem, should be determined so that the total energy dissipation rate reaches its minimum, i.e., $d\dot{E}/d\hat{x}=0$. The solutions obtained from $\hat{x}$ that satisfy this condition result in the minimum power dissipation ${\dot{E}}_{\min}$. Lastly, the deformation load *L* can be obtained as:(10)$$L=2{\dot{E}}_{\min}/U.$$

ACTA requires an analytical solution of $\hat{x}$, while ACTS can use either a closed-form solution or an optimization method to find the minimizing $\hat{x}$. Further details of these two modules are provided in the following sections.

### 4.2. ACT Simulator

ACTS takes the defined problem from the preprocessing stage, determines the zone of each field node, calculates the nodal velocities, and moves the nodes accordingly for an assigned increment. The problem should be set by defining the following items:(i)an admissible velocity field and the conditions for dynamically defining its internal boundaries (if exist),(ii)the geometrical domain of the initial sample in its pre-deformed shape, represented by field nodes,(iii)the ram speed *U*, which can be a constant or a function of other available variables,(iv)the initial distributions of the state variables of the defined constitutive law, such as stress and strain,(v)the constitutive law,(vi)the constant friction factor *m*,(vii)either the time step size Δt or the number of increments *N*, and(viii)the conditions for stopping the test, e.g., assigning a final height.

Regarding item (v), it is common practice to assume that the material obeys the von Mises yield criterion, which connects the yield stress in pure shear κ and normal loading σ via $\kappa =\sigma /\sqrt{3}$. This way, the shear stress in the frictional power can be written as a function of σ.

Once the problem is defined at the preprocessing stage and passed to the solver, the processing begins with position integration, that is, to find the position of each node moving according to the assigned velocity field. The new nodal positions are approximated using the forward Euler kinematics:(11)$$p^{\left(i+1\right)}=p^{\left(i\right)}+\Delta tU^{\left(i\right)},$$
where p and U are the nodal position and velocity, respectively, and the upper indices (i+1) and (i) indicate the corresponding time steps of $t^{\left(i+1\right)}$ and $t^{\left(i\right)}$, respectively, and $\Delta t=t^{\left(i+1\right)}-t^{\left(i\right)}$. Then, the nodal strain rate $\dot{\varepsilon }$ are calculated using Equation (Equation 8), which can be accomplished using either its closed-form solution, if available, or numerical methods.

The next step is to calculate the secondary nodal state variables required by the assigned constitutive law. For example, the nodal strain at the end of each increment is calculated by:(12)$$\varepsilon^{\left(i+1\right)}=\varepsilon^{\left(i\right)}+\Delta t{\dot{\varepsilon }}^{\left(i\right)}.$$

Once all relevant state variables have been obtained, the nodal stress values at any given increment are calculated using the predefined constitutive law. All of these secondary nodal properties (for example, ε and σ) are defined in terms of the unknown parameter(s) of the assigned velocity field and must be obtained as such that the minimizing condition of $d\dot{E}/d\hat{x}=0$ is satisfied. If a closed-form solution of $\hat{x}$ is at hand (${\hat{x}}_{\left(c\right)}$), it can be used directly to evaluate these properties. Otherwise, $\hat{x}$ can be found by performing minimization techniques (${\hat{x}}_{\left(\min\right)}$) such as the simplex search method of Lagarias et al. [29] based on the work of Nelder and Mead [30].

After ${\dot{E}}_{\min}$ is evaluated at the given increment *i* (that is ${\dot{E}}_{\min}^{\left(i\right)}$), the corresponding deformation load can be calculated from Equation (Equation 10). I should note that for the sake of brevity, the upper indices of (i) are omitted for the rest of the paper, but only for the increment (i); other time steps, such as (i+1), (0), or (f), are mentioned wherever required.

The final step of ACTS for increment *i* is to check whether the condition(s) to finish the simulation are satisfied: if not, it continues to the next increment; otherwise, the results are passed onto the postprocessing stage, where various data can be evaluated, namely, load-displacement curve, evolution of the sample’s profile, and distributions of state variables throughout the sample, e.g., stress and strain.

### 4.3. ACT Analyzer

The ACTA module proposed in this section is developed by assuming only two state variables in the constitutive law: stress and strain, which is the same as all available models in the literature; however, it is possible to extend it to capture other relevant state variables as well. ACTA requires six data sets for the preprocessing; the first four are the same as the ones for ACTS, and the other two are:(v)the load-displacement curve, and(vi)some geometry measurements of the sample at different time steps.

The employed velocity field, which is the core of ACTA, may require incremental measurements of the full sample’s profile R(z) or the slip radius $R_{S}$. With current test rigs, collecting such data is infeasible in a continuous test; therefore, the employed velocity field may necessitate performing a series of interrupted tests to use ACTA.

The objective of ACTA is to analyze the deformation load and the geometry of the sample as functions of displacement to obtain two results: (1) the average (and pointwise) flow curve of the sample and (2) the interfacial friction factor *m*. The following is my suggestion for implementing ACTA in a three-step method, which borrows two main assumptions from the CPM. The first is to approximate the average strain by $\bar{\varepsilon }\approx \ln\left(H_{0}/H\right)$. Furthermore, the corresponding stress distribution is approximated with a single average value, that is, $\bar{\sigma }=\sigma$; therefore, Equation (7) can be rewritten as:
(13a)$$\begin{array}{cc} {\dot{E}}_{d} & =2\pi \bar{\sigma }\int_{0}^{H/2}\int_{0}^{R(z)}\dot{\bar{\varepsilon }}rdrdz,\mathrm{and} \end{array}$$
(13b)$$\begin{array}{cc} {\dot{E}}_{f} & =m\frac{4\pi }{\sqrt{3}}\bar{\sigma }\int_{0}^{R_{T}}U_{r}\left(r,H/2\right)rdr. \end{array}$$

#### 4.3.1. Step 1: Estimating the Average Stress ($\tilde{\sigma }$)

Let us define a virtual optimal area ${\bar{A}}_{\mathrm{opt}}$ that connects the average stress and the compression load via:(14)$$\bar{\sigma }=L/{\bar{A}}_{\mathrm{opt}}.$$ Inserting Equations (Equation 10), (13) and (Equation 14) into Equation (Equation 5), the optimal area can be defined as:(15)$${\bar{A}}_{\mathrm{opt}}=\frac{2}{U}\left({\dot{E}}_{d}^{\prime }+{\dot{E}}_{f}^{\prime }\right),$$
where ${\dot{E}}_{d}^{\prime }$ and ${\dot{E}}_{f}^{\prime }$ are the deformation and frictional power normalized by $\bar{\sigma }$. ${\dot{E}}_{d}^{\prime }$ is a function of only the sample’s geometry, which is available at any increment. However, the calculation of ${\dot{E}}_{f}^{\prime }$ requires a priori knowledge of *m*, which is inaccessible in a physical setup. To calculate ${\bar{A}}_{\mathrm{opt}}$, we can identify its possible range by trying two limiting values of m={0,1}. Therefore, we find the lower (${\bar{A}}_{L}$) and upper (${\bar{A}}_{U}$) limits of ${\bar{A}}_{\mathrm{opt}}$. As for the first guess, let us approximate the optimal area as the mean of the two limits, i.e.,
(16)$${\tilde{A}}_{\mathrm{opt}}\approx \frac{{\bar{A}}_{L}+{\bar{A}}_{U}}{2}=\frac{2{\dot{E}}_{d}^{\prime }+{\dot{E}}_{f}^{\prime }\left(m=1\right)}{U}.$$ In this way, our first guess of the average stress is obtained as $\tilde{\sigma }=L/{\tilde{A}}_{\mathrm{opt}}$, assuming $\bar{\sigma }\approx \tilde{\sigma }$.

#### 4.3.2. Step 2: Estimating the Friction Factor ($m_{E}$)

The second step is to perform a series of ACTS by defining the constitutive law of $\tilde{\sigma }$-$\bar{\varepsilon }$ and by varying the friction factor in the range of [0,1]. Each of these ACTS results in a solution of $\hat{x}$ corresponding to their minimum power dissipation. Comparing the solutions of $\hat{x}$ obtained from a minimization process (${\hat{x}}_{\left(\min\right)}$) within the performed ACTS with those obtained from the closed-form solutions (${\hat{x}}_{\left(c\right)}$), the friction factor should be obtained from the one simulation that results in ${\hat{x}}_{\left(\min\right)}\approx {\hat{x}}_{\left(c\right)}$. The value of *m* used in that specific simulation would be picked as the estimated friction factor $m_{E}$.

#### 4.3.3. Step 3: Updating the Average Stress ($\bar{\sigma }$)

The average stress is then updated with the estimated friction factor. To do so, first, the optimal area in Equation (Equation 15) is updated by recalculating the normalized frictional power using $m_{E}$. Then, the average stress is updated using Equation (Equation 14).

## 5. Materials and Methods

In the following sections, two realizations of ACTAS are described and then benchmarked against comparable FE models. In all of these scenarios, the ACT is modeled as described in [11]: a cylindrical sample with initial height and radius of $H_{0}=16$ mm and $R_{0}=5$ mm is compressed with a uniform velocity of U=2 mm s^−1^ to its final height of $H_{f}=10$ mm. Moreover, due to the axisymmetrical description of the selected velocity fields, the test sample is modeled as only a cross-section of the cylindrical sample bounded in $r=[0,R_{0}]$ and $z=[0,H_{0}/2]$.

Different friction conditions are studied, assuming the friction factor being fixed throughout the test. Furthermore, the material is defined by Hollomon’s work-hardening constitutive law, that is, $\sigma =k\varepsilon^{n}$, where *k* is the strength coefficient and *n* is the strain hardening exponent [31]. It should be noted that while ACT plays an important role in mechanical characterization of materials during thermomechanical processes, which require advanced constitutive laws, e.g., [32], the simple material model of Hollomon serves the intent of the current study that is to introduce the new framework of ACTAS. In the current study, both *k* and *n* are assumed as material constants, varying in ranges of k∈[50,500]MPa and n∈[0,1]. The following are assumed for the development phases of ACTAS: m=0.3, k=100MPa and n=0.2.

FEM simulations are performed using DEFORM 2D with axisymmetric isoparametric four-node quadrilateral elements. For FE models, the samples are discretized with 2663 nodes and 2555 elements, with coarse elements near the *r* and *z* axes and fine ones close to the edge of the sample where foldover is expected the most. The FEM simulations are performed for 50 steps, and the deformation data are recorded every 10 steps, which resembles geometry measurements in a series of interrupted physical tests.

## 6. Creating a Virtual Laboratory Using ACTAS

This section demonstrates two practices (Cases 1 and 2) used for creating a virtual laboratory based on the proposed ACTAS framework. Case 1 describes a one-zone sample that is unable to show foldover. The main purpose of this case is to examine ACTAS’ abilities for a simple case. Case 2 introduces a more advanced velocity field that can capture the foldover phenomenon. In both cases, first the simulator (ACTS) and then the analyzer (ACTA) are described.

### 6.1. Case 1: A One-Zone Sample

#### 6.1.1. Velocity Field

ACTAS uses a velocity field (VF) at its core. Among numerous available options, the VF proposed by Kobayashi and Thomsen and modified by Lee and Altan is a simple yet promising choice [33,34]; hereafter, I refer to this VF as LAKT. In LAKT, the origin of the *z* axis is located in the middle of the sample and is defined as the following for the range of z∈[0,H/2]:
(17a)$$\begin{array}{cc} \; & U_{r}\left(r,z\right)=A\left(1-b_{L}z^{2}\right)r, \end{array}$$
(17b)$$\begin{array}{cc} \; & U_{z}\left(z\right)=-2A\left(z-b_{L}z^{3}/3\right), \end{array}$$
(17c)$$\begin{array}{cc} \; & U_{\varphi }=0, \end{array}$$
with
(18)$$A=\frac{U/2}{H(1-b_{L}H^{2}/12)}$$
determined using the velocity boundary condition of $U_{z}\left(H/2\right)=-U/2$. The arbitrary barreling parameter of the model $b_{L}$ represents the barreling of the deformed sample and can be estimated following the kinematic approach suggested in [12]. For a set of forward kinematics, the following holds:(19)$$\frac{R_{M}^{\left(i+1\right)}-R_{M}}{R_{T}^{\left(i+1\right)}-R_{T}}=\frac{U_{r}\left(R_{M},0\right)}{U_{r}\left(R_{T},H/2\right)}=\frac{R_{M}}{R_{T}\left(1-b_{L}H^{2}/4\right)}.$$ This analysis results in an estimate of the barreling parameter $b_{L(K)}$:(20)$$b_{L(K)}={\left(\frac{2}{H}\right)}^{2}\left(1-\left(\frac{R_{T}^{\left(i+1\right)}}{R_{T}}-1\right){\left(\frac{R_{M}^{\left(i+1\right)}}{R_{M}}-1\right)}^{-1}\right).$$ Moreover, the effective strain rate field of this velocity field can be formulated as [34]:(21)$$\dot{\bar{\varepsilon }}=\frac{2A}{\sqrt{3}}\sqrt{3{\left(1-b_{L}z^{2}\right)}^{2}+{\left(b_{L}rz\right)}^{2}}.$$

LAKT has a few features that makes it an excellent start for ACTAS implementation: it is simple and easy to use, and its profile description is comparable with experiments [34], unlike those models with exponential functions. Moreover, although LAKT leads to no foldover, its radial velocity component being a function of both *r* and *z*, is crucial to extend it for capturing the foldover phenomenon. More details are provided in Section 6.2.

#### 6.1.2. ACT Simulator

The implementation of ACTS using LAKT in the current study is almost identical to the one described by Lee and Altan [34]. The ACT is simulated in a number of steps *s*, directly affecting the incremental height reduction: $\Delta h=(H_{0}-H_{f})/s$. At the beginning of each step, the Nelder–Mead simplex algorithm [29,30] is used to minimize $\dot{E}$ with respect to $b_{L}$. Then, the nodal velocity field, strain rate, strain, and stress are calculated in order. Then, the simulation process continues to the next step unless it satisfies one of the following conditions: (1) the final height ($H_{f}$) is reached or (2) the radius of the top plane $R_{T}$ shrinks. The latter is a physically unjustifiable artifact that may occur because of the features of the velocity field. In such a case, the further solutions of the model are invalid.

The implemented ACTS outputs the final shape of the sample, the distributions of radial and vertical components of the velocity field, strain rate, strain, and stress. In addition, it records the deformation load and the barreling parameter at each increment.

As for the first step and before performing the primary investigations, the appropriate number of time steps and sample nodes must be decided. For that, various time steps of {5, 10, 25, 50, 75, 100, 250, 500, 750, 1000, 2500, 5000, 7500, 10,000} and grids of {3, 5, 10, 30, 50, 100} are studied. Here, the number of grids refers to the nodes along each axis; for example, a grid number of 3 means that the sample was defined by three sets of nodes along the *r* axis and three sets of nodes along the *z* axis, i.e., 3×3=9 nodes in total. To measure the accuracy of each simulation, the volume of the sample in its initial ($V_{0}$) and final ($V_{f}$) stages is used to calculate a percentage volume change error by $\delta \left(V\right)=100\%\times \left|1-V_{f}/V_{0}\right|$, where |…| indicates the absolute value. Figure 3 shows the results of these tests. Choosing an arbitrarily low-value threshold of $\delta {\left(V\right)}_{\max}=0.02\%$, the set with the time step 1000 and the grid number 50 is selected for all forthcoming studies in the current work.

With the selected set of time steps and grids, ACTS is performed on the sample. Figure 4 shows the discretized sample at the initial and final stages. Changes in the radial profile are tracked during deformation. Figure 5a shows the mid-plane $R_{M}$ and top-plane $R_{T}$ radii. Moreover, Figure 5a shows the calculated deformation load *L*. Figure 5b shows two sets of barreling parameters. One is obtained from the minimization process of the dissipation energy ($b_{L(\min)}$), and the other is estimated from the kinematic analysis of LAKT ($b_{L(K)}$ from Equation (Equation 20)) using the recorded geometry of the deformed sample. Figure 5b shows that the two methods result in the same solutions of $b_{L}$.

#### 6.1.3. ACT Analyzer

To examine the suggested ACTA framework, 100 tests are analyzed by collecting their incremental data of heights, deformation load, mid-, and top-plane radii. The sampling is performed by varying the friction factor (*m*) and the material parameters (*k* and *n*) using the Latin Hypercube Sampling (LHS) algorithm, and the data collection is performed from ACTS virtual experiments.

To assess the performance of ACTA, two measures are used: the mean absolute relative error (MARE) and the coefficient of determination ($r^{2}$), which are defined as:
(22a)$$\begin{array}{cc} \; & MARE=\frac{1}{N}\sum_{j}|\frac{{X_{A}}_{\left(j\right)}-{X_{E}}_{\left(j\right)}}{{X_{A}}_{\left(j\right)}}|,\mathrm{and} \end{array}$$
(22b)$$\begin{array}{cc} \; & r^{2}=1-\frac{\sum_{j}{\left({X_{A}}_{\left(j\right)}-{X_{E}}_{\left(j\right)}\right)}^{2}}{\sum_{j}{\left({X_{A}}_{\left(j\right)}-{\bar{X}}_{A}\right)}^{2}}, \end{array}$$
where *X* is the selected observable (*m*, *k*, or *n*), *N* is the total number of observations with *j* pointing to a single observation, with ${\bar{X}}_{A}=\left(\sum_{j}{X_{A}}_{\left(j\right)}\right)/N$. The indices A and E refer to the assigned and estimated values, respectively. A perfect match can be identified by the lower bound of MARE=0 and the upper bound of $r^{2}=1$.

The estimated friction factor is the product of ACTA’s step 2. Figure 6 shows the correlation between the assigned ($m_{A}$) and estimated ($m_{E}$) values of the friction factor. The results show a good correlation between $m_{A}$ and $m_{E}$ up to a maximum value of ∼0.6. Although larger values of $m_{A}$ are examined, those simulations satisfy the stopping condition 2, that isidentifying a shrinkage in $R_{T}$, and are rejected.

The average stresses are calculated by taking the estimated friction factors into the third step of ACTA. Together with the corresponding average strains, these values established the average stress-strain curves. To compare all estimated constitutive models with the assigned ones, they are fitted with the Hollomon model due to the a priori knowledge about their form. Figure 7 shows the correlation between the assigned and estimated values of *k* and *n* for all tests.

The analysis results on both the friction factor and the material parameters reveal a high accuracy of the proposed ACTA framework.

### 6.2. Case 2: Capturing the Foldover of the Side Surface

To model the foldover phenomenon, the model should be able to solve two issues. The first one is about the ability of the velocity field to treat the sample in more than one zone. The other issue is the dynamic evolution of the sample, so that the nodes on the side surface can join the top layer upon foldover and contribute to the expansion of the top-plane radius from $R_{S}$ to $R_{T}$.

#### 6.2.1. Velocity Field

To capture the foldover phenomenon, the selected velocity field must divide the sample into more than one zone. One can use advanced velocity fields from Dadras and Thomas [25] or Hou et al. [26], each with four-zone models. However, to examine the ACTAS framework, a simpler velocity field with two deformation zones may be sufficient. Therefore, I propose a two-zone model as shown in Figure 8, with zone 1 for $r\leq R_{T}$ and zone 2 for $r>R_{T}$ that is the bulging part.

The downward material flow in zone 2 (unlike in zone 1) is not directly controlled by the boundary contact of the compressing tool. In this two-zone model, LAKT is used for zone 1. For zone 2, the proposed velocity field of Hou et al. [26] of a comparable zone is used, with zero vertical movement. Thus, the new two-zone velocity field can be read as follows:(23)$$U_{z}\left(r,z\right)=\left\{\begin{array}{cc} -2A\left(z-b_{\lambda }z^{3}/3\right) & \mathrm{if}\;r\leq R_{T}\\ 0 & \mathrm{if}\;r>R_{T} \end{array}\right.,$$
where *A* can be easily found to be:(24)$$A=\frac{U/2}{H(1-b_{\lambda }H^{2}/12)}.$$ The radial velocity component in each zone is then obtained from the integration of the incompressibility equation given by:(25)$$\frac{\partial U_{r}}{\partial r}+\frac{U_{r}}{r}+\frac{\partial U_{z}}{\partial z}=0,$$
resulting in:(26)$$U_{r}\left(r,z\right)=A\left(1-b_{\lambda }z^{2}\right)\times \left\{\begin{array}{cc} r & \mathrm{if}\;r\leq R_{T}\\ \left(R_{T}^{2}/r\right) & \mathrm{if}\;r>R_{T} \end{array}\right..$$

The strain rate components are obtained from the velocity field components, and the effective strain rate is formulated as:(27)$$\dot{\bar{\varepsilon }}=\frac{2A}{\sqrt{3}}\times \left\{\begin{array}{cc} \sqrt{3{\left(1-b_{\lambda }z^{2}\right)}^{2}+{\left(b_{\lambda }rz\right)}^{2}} & \mathrm{if}\;r\leq R_{T}\\ \sqrt{\;\;{\left(1-b_{\lambda }z^{2}\right)}^{2}+{\left(b_{\lambda }rz\right)}^{2}}\left(R_{T}^{2}/r\right) & \mathrm{if}\;r>R_{T} \end{array}\right..$$

Moreover, following the proposed method in [12] for a forward kinematic estimation of the barreling parameter, $b_{\lambda }\left(K\right)$ for this two-zone velocity field is obtained as:(28)$$b_{\lambda }\left(K\right)={\left(\frac{2}{H}\right)}^{2}\left(1-\frac{\left({R_{□}}^{\left(i+1\right)}-R_{□}\right)R_{□}}{\left(R_{M}^{\left(i+1\right)}-R_{M}\right)R_{M}}\right).$$ Note that $R_{T}$ is replaced by $R_{□}$ because of the incapability of the two-zone model to address foldover in its closed-form formulation. The initial zone 1 expands only to $R_{S}$, and the foldover contributes to the final measured $R_{T}$. Therefore, $R_{T}$ becomes increasingly larger than the predictions of the model, resulting in underestimates of $b_{\lambda }$. The solutions of ACTS are are used to decide on the values of $R_{□}$.

#### 6.2.2. ACT Simulator

The method for implementing ACTS with the new velocity field is more or less the same as described for Case 1 (Section 6.1.2), with no difference in the setup. Upon the beginning of the simulation process, all nodes are considered to be in zone 1. As soon as $R_{M}$ becomes larger than $R_{T}$, the nodes in zone 2 are flagged. Then, once any of the nodes in zone 2 reaches the height of the compressing tool, its flag is removed, and the node is considered as a field node in zone 1. This node, traveling from zone 2 to zone 1, represents the foldover. Therefore, the top-plane radius $R_{T}$ is updated using the radial position of this field node.

Figure 9 shows the discretized sample at its final stage of the simulated ACT, exhibiting dynamically identified slip ($R_{S}$) and top-plane ($R_{T}$) radii, with their difference indicating the identified foldover. The percentage volume change error of the simulation was calculated to be a negligible value of δ(V)≈0.03%.

Figure 10a shows the selected radii and the deformation load. Comparing the results with Figure 5a (Case 1), it is noticeable that the changes of $R_{S}$ in Case 2 are close to the changes of $R_{T}$ in Case 1, which can be justified with the fact that both are results of LAKT. The changes of $R_{M}$ and $R_{T}$ are different between the two cases. In Case 2, where the foldover is captured, $R_{M}$ increases slower than in Case 1, and the opposite is true for $R_{T}$. The incompressibility assumption justifies this behavior. The deformation loads for both cases are more or less the same. Furthermore, Figure 10b shows the minimized and kinematic estimations (Equation (Equation 28)) of the barreling parameter $b_{\lambda }$. Two values of $R_{S}$ and $R_{T}$ are tried for $R_{□}$. The results show that $R_{T}$ results in large deviations as soon as the foldover initiates. Using $R_{S}$ results in some discrepancy as deformation advances and foldover becomes more severe; however, the deviations are small. Based on these expected results, it is assumed that $R_{□}=R_{S}$ for the rest of this investigation. Although this does not guarantee a perfect match between $b_{\lambda }\left(\min\right)$ and $b_{\lambda }\left(K\right)$ for the whole range of deformation, their values are close enough for a wide range with minimal differences.

#### 6.2.3. ACT Analyzer

A new set of 100 samples is prepared using the LHS algorithm and tested as follows. First, virtual compression tests are performed using ACTS on all samples. Then, using their collected data, they are analyzed using ACTA, without any further modifications to its algorithm.

Figure 11 shows the correlation between the assigned and estimated friction factors obtained from ACTA’s step 2. Compared to Case 1 (Figure 6), the results show that the newly proposed model enables ACTS to perform on a full range of m∈[0,1]. Figure 12 shows the mechanical characterization results using ACTA, with the correlations between the assigned and estimated material parameters of *k* and *n* for all tests.

The calculated errors of the results obtained for both the friction factors and the material parameters indicate the applicability of ACTA for highly accurate estimates.

## 7. Benchmarking ACTAS against FEM Virtual Experiments

In a physical setup, the flow stress of the material is unknown and must be obtained from the interpretation of the recorded load-displacement data. However, the flow stress curve and the frictional conditions are a priori knowledge in a virtual simulation. This difference makes virtual tests superior to their physical counterparts in assessing an analyzer. Here, FEM-based tests are considered reference virtual experiments.

Moreover, several geometrical data as functions of the sample’s height are required to work with any flow-based model of ACT, such as the two-zone VF proposed in the current investigation. Currently, available compression test rigs cannot perform such measurements [1]. As a remedy, interrupted tests may need to be executed. Here, to simulate interrupted tests, the required geometric data ($R_{S}$, $R_{T}$, and $R_{M}$) are collected only five times during each virtual experiment.

Ten samples are prepared using the LHS algorithm; Table 1 summarizes their characteristics. These samples benchmark the proposed ACTAS (based on the method discussed in Case 2), and the corresponding FEM solutions are considered references. The assessment of ACTAS is performed using different error measurements. For the multi-valued solutions of ACTS, i.e., load and geometrical measurements as functions of displacement, the absolute percentage error of each quantity for the whole course of ACT is calculated via:(29)$$\delta \left(X\right)=100\%\times \left|1-\frac{X_{\mathrm{ACTS}}}{X_{\mathrm{FEM}}}\right|,$$
where *X* is the quantity under study. For the single-valued solutions of ACTA, i.e., *m*, *n*, and *k*, the same errors of MARE and $r^{2}$ (Equation (22)) are considered.

### 7.1. ACT Simulator

Figure 13 shows the comparison between the deformation load and geometry of the deformed samples for the two direct solutions. The maximum error of the deformation load is around 6%. For geometry measurements, the errors are found to be negligible in general, except for sample 10 with the highest friction factor.

### 7.2. ACT Analyzer

The data collected from the FE models are analyzed using the ACTA framework to estimate the sample’s stress-strain curve and the interfacial friction factor in each test. Figure 14 compares the assigned and estimated friction factors. Moreover, the flow stress curves are obtained and Hollomon’s model is fitted to them (as justified earlier in Section 6.1.3). Figure 15 compares the fitted and assigned constants of the models. The results show that all estimated values are closely comparable with the assigned ones.

## 8. Results and Discussion

ACT is a powerful tool in characterizing metals’ hot and cold flow behaviors and their associated phenomena, e.g., recovery and recrystallization. For this purpose, it is common practice to perform microstructural analyses on the center core of the sample. Despite the microstructural analyses’ locality, the corresponding flow curves are obtained with the assumption of uniform distribution of stress and strain throughout the sample. In this section, the benchmark results are discussed further and compared with the results of the conventional methods.

### 8.1. Average Stress

Figure 16 shows the stress-strain curves obtained from the analyses of the FE models using ACTA and A-CPM (Equation (Equation 4)) in comparison with the assigned ones. The two estimations are not precisely the same for the whole strain range of all samples; however, their estimation errors (for all collected data points) are almost the same as each other, with $r^{2}=0.998$ and MARE=0.03, indicating high accuracy for both analysis methods. The main difference between the two is their required computational power: ACTA is an incremental numerical method that requires many calculations, while A-CPM (Equation (Equation 4)) is a closed-form solution with one single calculation per data point. For this reason, A-CPM is the most efficient of the two. Yet, one should note that A-CPM works here only due to the small contribution of foldover in the average behavior of the samples.

To employ A-CPM, the barreling parameter of the Avitzur model $b_{A}$ is needed. See Appendix A for detailed study on the various kinematic estimates of $b_{A}$ proposed in [12]. Among all estimates of $b_{A}$ in [12], those developed for a static method can cover a wider range because they provide only one value of $b_{A}$ for the entire course of deformation. In contrast, the ones for a dynamic method identify one value for each geometry measurement. Although for a reliable conversion of the force-displacement data into a flow stress curve, one should use the formulae developed within the dynamic method, these formulae limit the number of the converted data points to the number of geometry measurements during the test. For example, in the current work, the profile is measured five times during the simulated ACT, and the dynamic solutions provide only five data points on the flow curve. To convert the whole range of deformation, the formulae developed within the static method could be used, although they are prone to misleading conclusions, such as false identification of peak stress. Among the developed estimates of $b_{A}$, the two proposed initiallly by Ebrahimi and Najafizadeh [10] and Solhjoo [11] are the only ones that do not result in false interpretations for the studied samples (see Appendix A), both showing the same high levels of accuracy and precision. Of these two models, the one developed by Solhjoo is selected here, that is [11]:(30)$$b_{A(K)}\approx 4\frac{\Delta R}{\Sigma R}\left(\frac{2H}{\Delta H}-1\right),$$
with $\Delta R=R_{M}^{\left(f\right)}-R_{T}^{\left(f\right)}$, $\Sigma R=R_{M}^{\left(f\right)}+R_{T}^{\left(f\right)}$, and $\Delta H=H_{0}-H^{\left(f\right)}$, where the superscript (f) indicates the final state of the deformation.

### 8.2. Stress Distribution throughout the Sample

ACTAS is capable of providing the distributions of the state variables available within the model; for the current implementation, these state variables are strain rate, strain, and stress. Figure 17 compares the stress distribution of (randomly selected) sample 5. The solutions may seem to differ a lot; however, they are close at a crucial point: the center of the sample, where ACTA, ACTS, and FE models predict stresses of 218 MPa, 220 MPa, and 233 MPa, respectively. All of these predictions are higher than the average stress of ∼200 MPa of the sample; see Figure 16.

Table 2 summarizes the stress magnitudes at the center of the samples for the reference FEM experiments, compared with the ACTS, ACTA, and A-CPM solutions for all ten samples. The results show that all methods, the simulator (ACTS) and the analyzers (ACTA and A-CPM), underestimate the stress values; however, the overall percentage errors for ACTS and ACTA are reduced by ∼50% compared to A-CPM.

### 8.3. Strain Distributions

A-CPM defines uniform distributions for the state variables throughout the sample, which is valid for identifying the constitutive behavior of the material as a whole (Figure 16). However, the local values are requested for microstructural analysis of samples as they can be far from the average values. Such a deviation can also be present for the strain distribution. Figure 18 shows the strain distributions of the randomly selected sample 5 obtained from the FE, ACTS, and ACTA models. Despite the essentially different strain distribution patterns, the solutions toward the core of the sample are comparable. The strains at the center of all ten samples are summarized in Table 3, showing an overall improvement of ~60% in the prediction of the strain using ACTS and ACTA compared to A-CPM; ${\bar{\varepsilon }}_{A-\mathrm{CPM}}=0.47$ for all samples.

### 8.4. Flow Behavior at the Center of the Sample

For a clear comparison, the local stress-strain curves at the center of all ten samples are collected; Figure 19 shows these local flow curves. (Strain rates are not studied simply because the selected constitutive model is strain rate independent.) In all cases, the ACTS, ACTA, and A-CPM models underestimate both the stress and strain of the final data point of the FE models; however, as discussed earlier, ACTS and ACTA give better estimates (Table 2 and Table 3). The solutions of ACTS models coincide with those of the FE models for their entire identified (yet underestimated) ranges.

Moreover, the general shape of A-CPM can be misleading. For example, for sample 6 (Figure 19f), A-CPM suggests the flow stress is almost reaching its maximum, although the stress of the FE model is only increasing. One should note that the reported results of A-CPM are based on the best estimates of $b_{A}$, and other available solutions of $b_{A}$ could result in misleading solutions for other samples, too; see Appendix A for further details.

### 8.5. Friction Factor

Although ACTA is developed primarily to serve as a reliable interpreter of ACT and to characterize the mechanical behavior of materials, it can estimate the friction factor at the tool-sample interface. Table 4 summarizes the friction factors approximated with ACTA (Figure 14) and the Avitzur model (Equation (Equation 3)), and compares them with the assigned values. The results show the high accuracy of the proposed analyzer (ACTA) in estimating the friction factor. Moreover, the results confirm previous works on the inaccuracy and unreliability of estimating the friction factor using the Avitzur model.

One may argue that developing a method valid for an exceptionally high friction factor of m=1 is redundant, as the lubrication assures low friction during metal forming processes. While this might be true for the upper bound, friction factor can be evaluated experimentally only at its limiting values: m=0 for a fully lubricated process (if $R_{T}=R_{M}$) and m=1 for a sticking one (if $R_{T}=R_{0}$), and no other intermediate values, which is the case for most of the processes, can be identified by direct measurements. Therefore, a reliable analysis tool must be valid for a full range of *m* to be used safely for post-processing the measurements to yield reliable mechanical properties of the material.

## 9. Summary and Conclusions

This paper proposes a meshfree-based numerical framework (called ACTAS) that models the axisymmetric compression test in an incremental approach and uses the upper bound theorem to solve for the test’s deformation load, with a velocity field on its core to describe the sample’s material flow during the test. ACTAS consists of two modules: an analyzer (ACTA) and a simulator (ACTS). ACTAS is a general framework that can take various velocity fields to model the material flow of the ACT sample. If the implemented velocity field allows, ACTAS models the side surface foldover phenomenon.

The steps to build a virtual laboratory based on ACTAS for mechanical and frictional analyses of ACT are described in the paper. For this goal, two velocity fields are used in the implementation of ACTAS, which is then benchmarked against the reference models obtained from FEM-based virtual experiments. The following summarizes the main points.
To employ the ACTA module, closed-form solutions for the unknowns of the model must be available, e.g., the barreling parameters of the velocity fields. Due to the formulation of VFs, the derivation of the closed-form solutions may not be trivial or possible. Instead, one can use kinematics of the VF to estimate the solution of its unknowns; see, e.g., Equation (Equation 19).As the first step in building a virtual laboratory based on ACTAS, a one-zone VF (LAKT) [33,34] is used. The setup is selected such that the percentage volume change error becomes a negligible value of δ(V)≈0.02%. In this setup, ACTA correctly estimates all samples’ constitutive behavior and friction factors for m≲0.6, which is found to be the upper limit of LAKT to meaningfully model ACT.Due to the incapability of LAKT in modeling the foldover phenomenon, an extension of it is proposed; see Section 6.2 for the details. By implementing the newly proposed two-zone velocity field, the ACTA module obtains accurate results for the full range of 0≤m≤1.ACTAS is benchmarked against ten reference solutions obtained from FEM-based experiments (Table 1). The ACTS module shows low percentage errors for the deformation load (δ(L)≲6%) and geometrical measurements (Figure 13). The ACTA module accurately estimated the average stress-strain curves for all samples.Investigating the pointwise stress-strain curves at the center of the samples between ACTAS and FE models, ACTAS provides improved results compared to those of the conventional methods. Moreover, ACTAS results in no false identification of peak stress, misleading to interpretations about the onset of dynamic recrystallization.This paper also addresses the shortcoming of employing the CPM without a priori knowledge of friction. As a solution, the Avitzur model is coupled with the CPM (called A-CPM) and is represented as:
$$\frac{p_{\mathrm{ave}}}{\bar{\sigma }}\approx 1+{\left(\frac{\bar{R}}{H}\right)}^{2}{\left(\frac{6}{b_{A}}-1\right)}^{-1},$$
with $\bar{R}=R_{0}\sqrt{H_{0}/H}$ and $b_{A}$ being the barreling parameter of the Avitzur model, which can be estimated by geometrical measurements of the sample via, e.g., Equation (A2); see Appendix A for the details.Comparing the solutions of A-CPM and FE models, it is found that A-CPM can be used to accurately identify the friction-free average stress-strain curves regardless of the severity of friction.Because the solutions of A-CPM are almost identical to those of the Avitzur model (Figure 2), it infers the high accuracy of the Avitzur model in estimating the average stress-strain curves.The unreliability of the Avitzur model for estimating friction factor is confirmed once more, aligned with previous investigations.For microstructural analyses, the study is usually focused on the center of the sample, where the flow curve may differ from the average one. For such studies, the analyses of A-CPM should be considered with extra care, as it underestimates both the stress and the strain values. It may even provide misleading results regarding the onset of dynamic recrystallization.

ACTAS shows promising results and versatile features for implementing various velocity fields and constitutive behaviors. In this paper, which primarily aims to introduce this new framework and demonstrate its capabilities, simple models are selected for both options: a two-zone velocity field to describe the sample’s flow and the Hollomon’s constitutive law that defines stress to be only a functions of strain. ACTAS must be validated for more challenging material models, e.g., with strain rate and temperature dependence, to prove its robustness as a reliable ACT simulator and analyzer.

## Figures and Tables

**Figure 1 materials-16-00441-f001:**
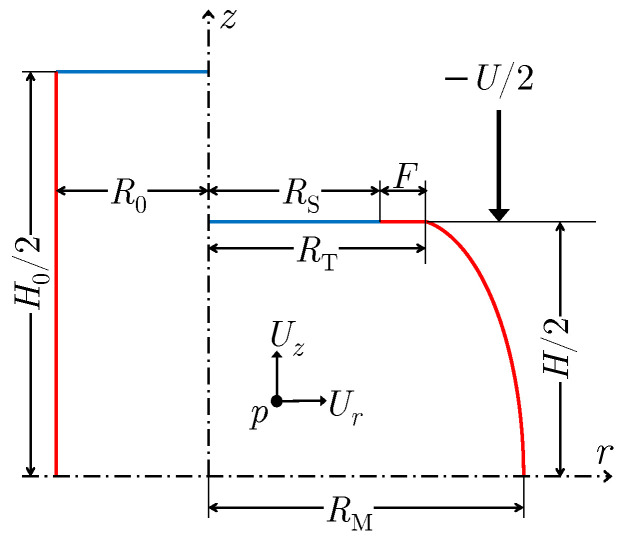
The schematic of the sample geometry in axisymmetric compression test (left) before and (right) after the deformation. $R_{0}$ and $H_{0}$ are the sample’s initial radius and height, respectively. By deforming the sample with a constant velocity of −U/2 parallel to the *z* axis, three characteristic radii can be identified: the mid-plane ($R_{M}$), top-plane ($R_{T}$) and slip ($R_{S}$) radii with foldover being $F=R_{T}-R_{S}$. ($R_{S}$ is the result of the expansion of $R_{0}$, and *F* is the contribution of the side surface foldover, resulting in a larger top-plane radius $R_{T}$). In a flow-based model, any point p(r,z) moves based on the radial and axial components of the velocity field ($U_{r},U_{z}$).

**Figure 2 materials-16-00441-f002:**
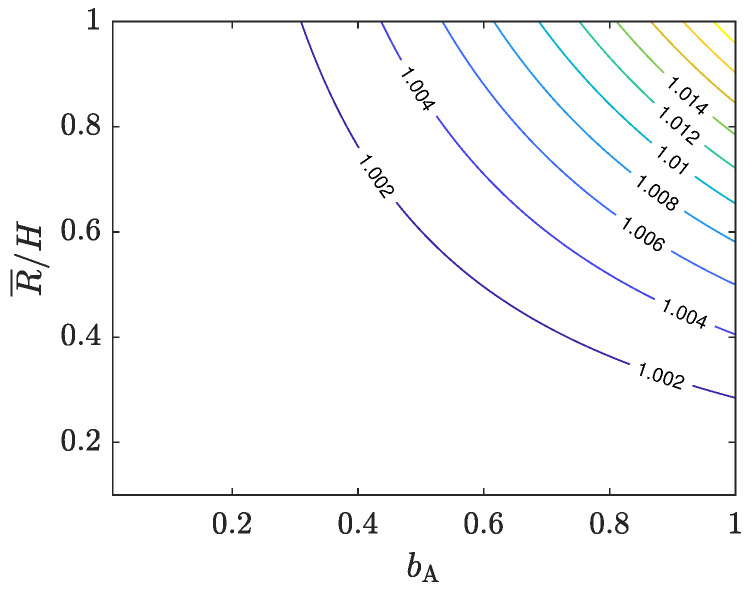
The ratio of Equation (Equation 4) (A-CPM) to Equation (Equation 2) (Avitzur model) for a wide range of $b_{A}$ and $\bar{R}/H$.

**Figure 3 materials-16-00441-f003:**
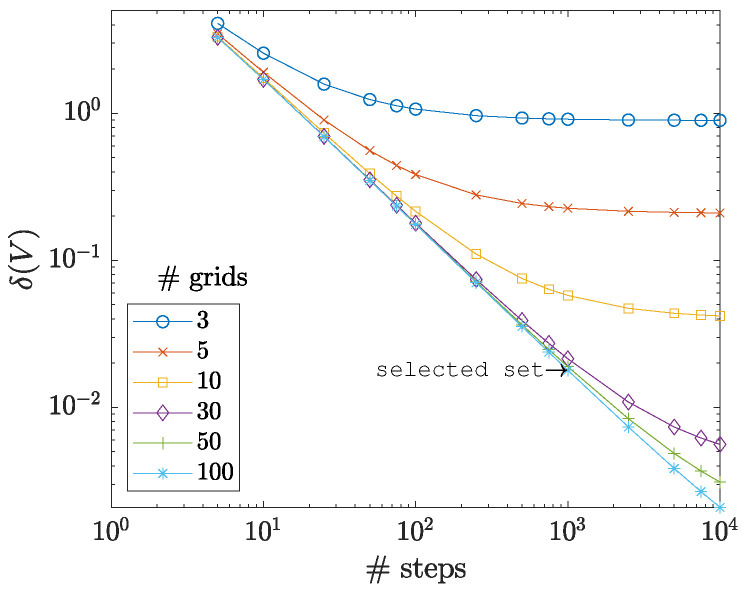
The percentage volume change error as a function of numbers of simulation steps and grids. With an arbitrarily small threshold of $\delta {\left(V\right)}_{\max}=0.02\%$, the features for the forthcoming tests of the current study are selected.

**Figure 4 materials-16-00441-f004:**
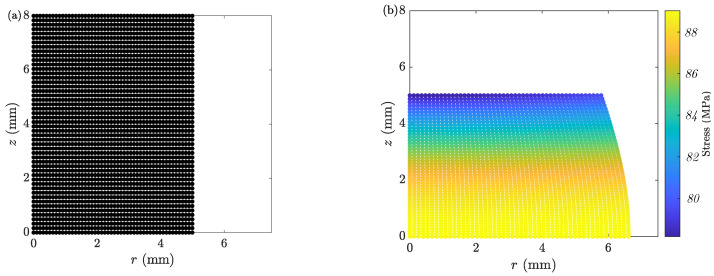
Discretized sample of Case 1 with a grid of 50 at the (**a**) initial and (**b**) final stage of the ACT.

**Figure 5 materials-16-00441-f005:**
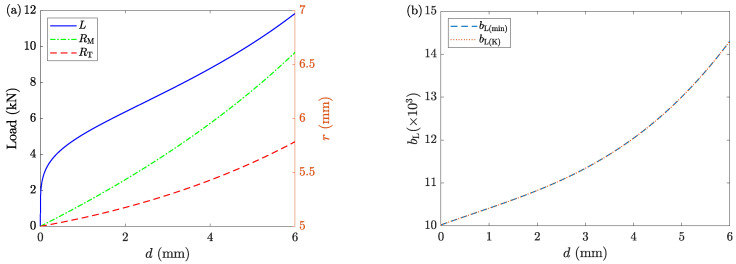
The results of Case 1. (**a**) The variation of deformation load (left *y*-axis), mid-plane, and top-plane radii (right *y*-axis) as functions of displacement *d*. (**b**) The values of $b_{L}$ obtained from the minimization process ($b_{L(\min)}$) and the kinematics estimation ($b_{L(K)}$) from Equation (Equation 20).

**Figure 6 materials-16-00441-f006:**
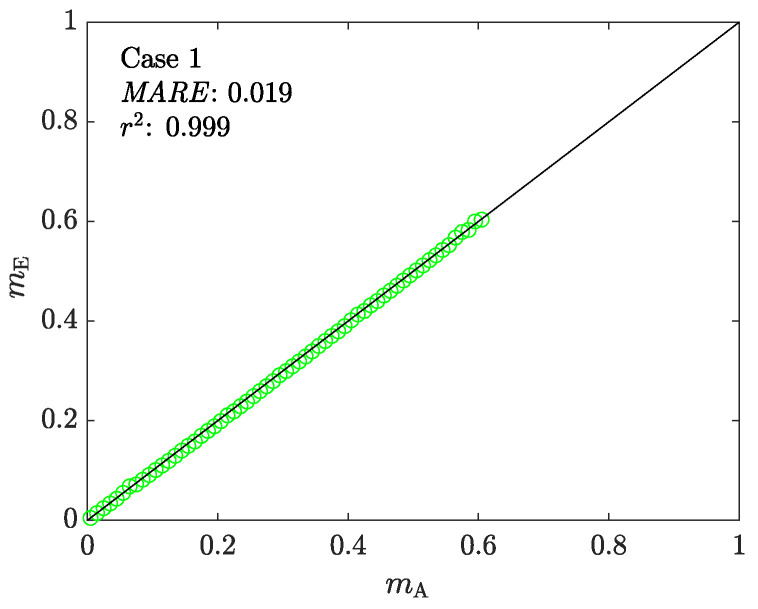
The correlation between the assigned ($m_{A}$) and estimated ($m_{E}$) values of the friction factor in Case 1. The continuous black line represents a perfect correlation.

**Figure 7 materials-16-00441-f007:**
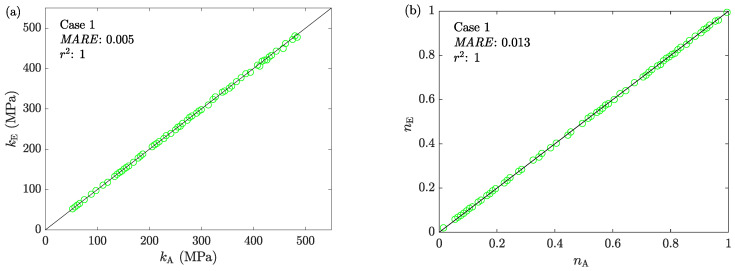
The correlation between the assigned and estimated values of the material parameters (**a**) *k* and (**b**) *n* in Case 1.

**Figure 8 materials-16-00441-f008:**
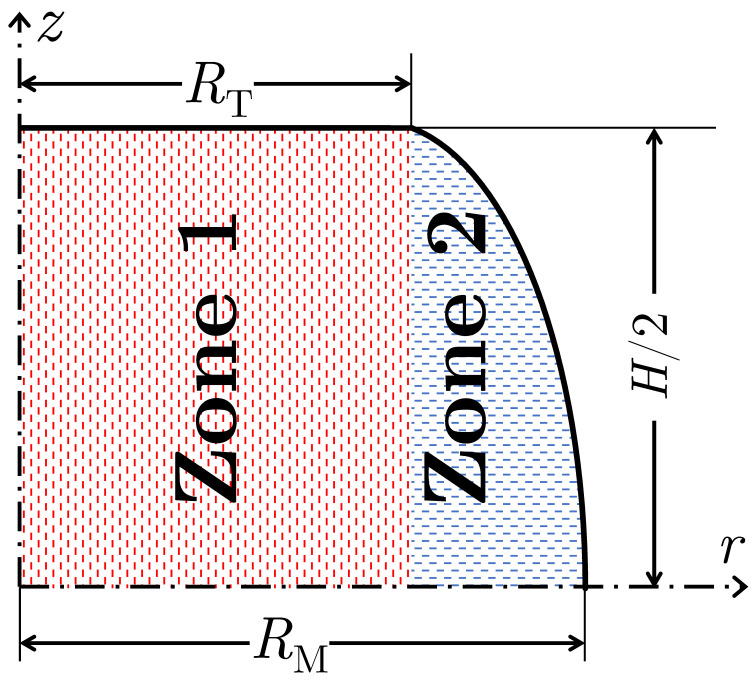
The schematic of the newly proposed model that divides the sample into two zones, with zone 1 for $0\leq r\leq R_{T}$ and zone 2 for $R_{T}<r\leq R\left(z\right)$. The border of zones 1 and 2 is at $r=R_{T}$.

**Figure 9 materials-16-00441-f009:**
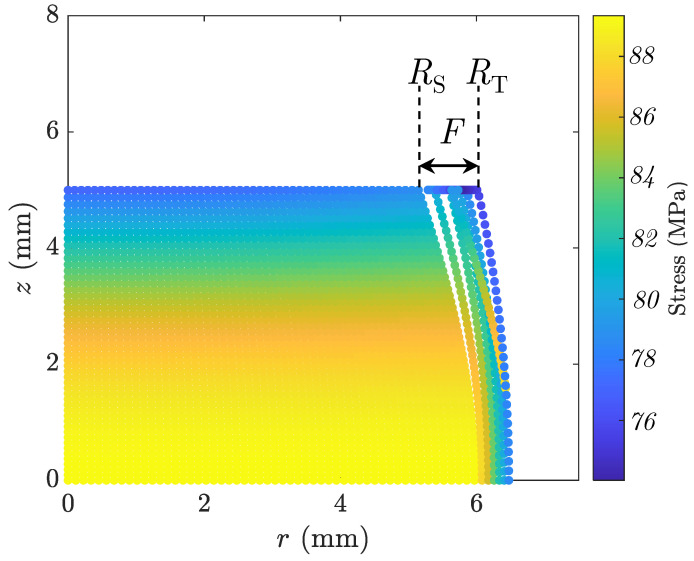
Discretized sample of Case 2 with a grid of 50 at the final stage of the ACT.

**Figure 10 materials-16-00441-f010:**
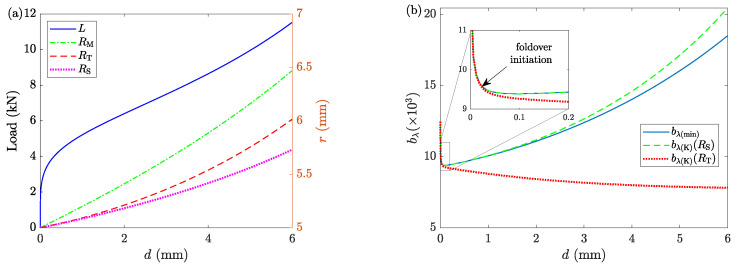
The results of Case 2. (**a**) The deformation load (left *y*-axis) and the sample’s geometry (right *y*-axis) as functions of displacement *d*. (**b**) The values of $b_{\lambda }$ obtained from the minimization process ($b_{\lambda }\left(\min\right)$) and the kinematics estimation ($b_{\lambda }\left(K\right)$) from Equation (Equation 28) for two values of $R_{□}$: $R_{S}$ and $R_{T}$. The zoomed-in inset shows that the $b_{\lambda }\left(K\right)\left(R_{T}\right)$ underestimates the barreling parameter from the onset of the foldover phenomenon.

**Figure 11 materials-16-00441-f011:**
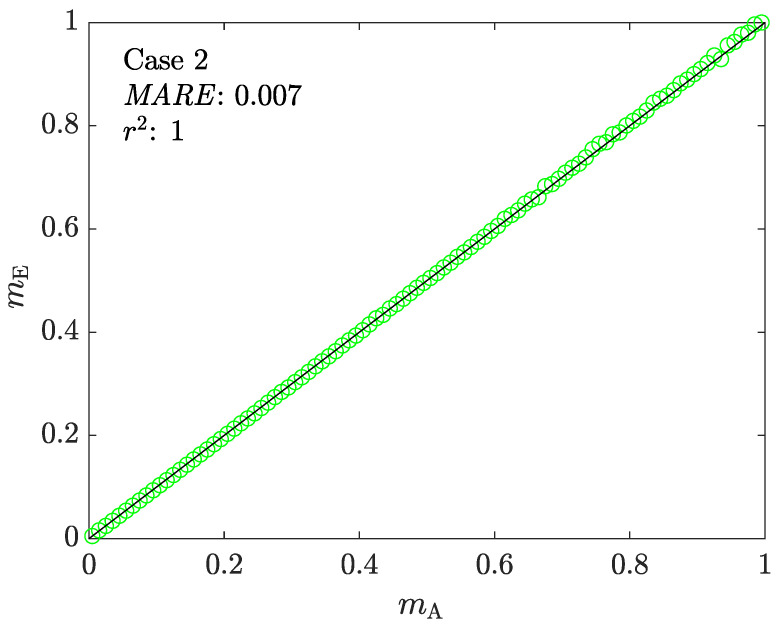
The correlation between $m_{A}$ and $m_{E}$ in Case 2.

**Figure 12 materials-16-00441-f012:**
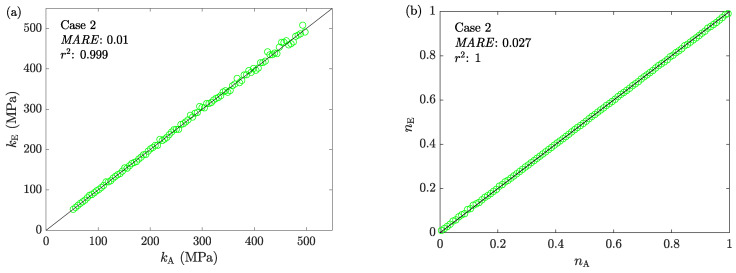
The correlation between the assigned and estimated values of the material parameters (**a**) *k* and (**b**) *n* for Case 2.

**Figure 13 materials-16-00441-f013:**
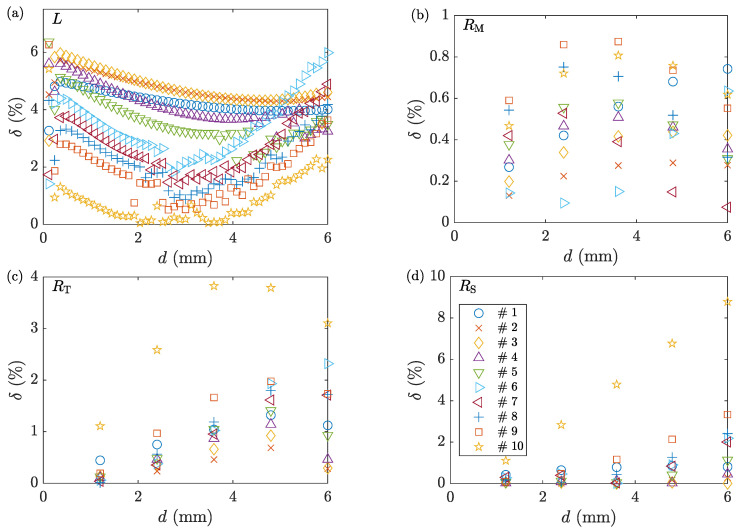
The percent error of (**a**) deformation load, and profile’s (**b**) $R_{M}$, (**c**) $R_{T}$, and (**d**) $R_{S}$ obtained from comparing ACTS and FE models. The symbols are to be read from the legend in the subplot (**d**) that refers to the corresponding sample number in Table 1.

**Figure 14 materials-16-00441-f014:**
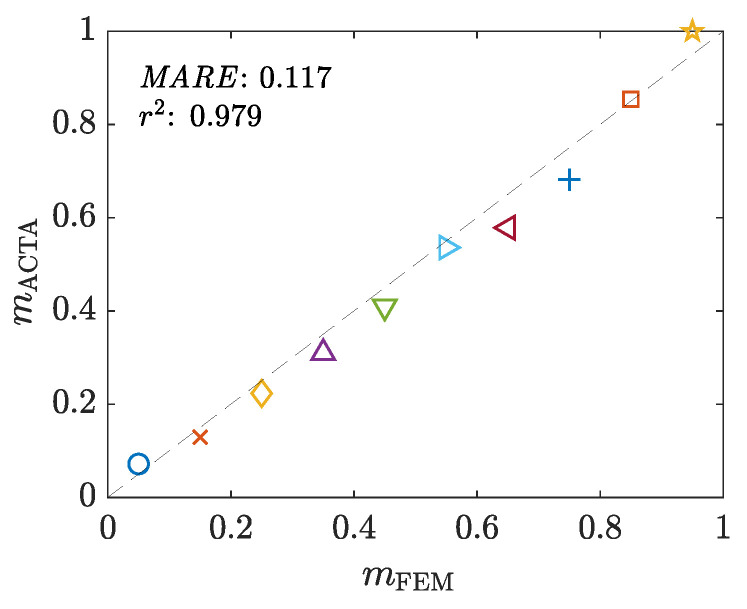
The correlation between the assigned ($m_{\mathrm{FEM}}$) and estimated ($m_{\mathrm{ACTA}}$) friction factors. The symbols point to different samples to be read using the legend of Figure 13d.

**Figure 15 materials-16-00441-f015:**
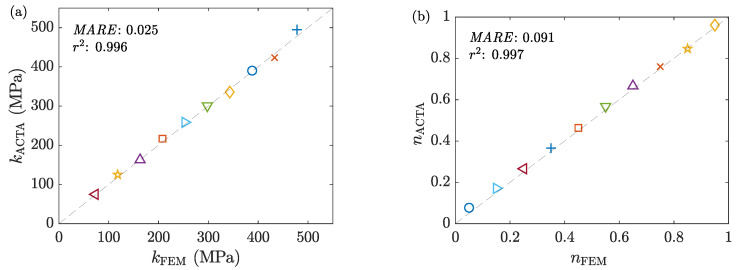
The correlation between the assigned and estimated values of the material parameters (**a**) *k* and (**b**) *n* for the samples investigated in the benchmark.

**Figure 16 materials-16-00441-f016:**
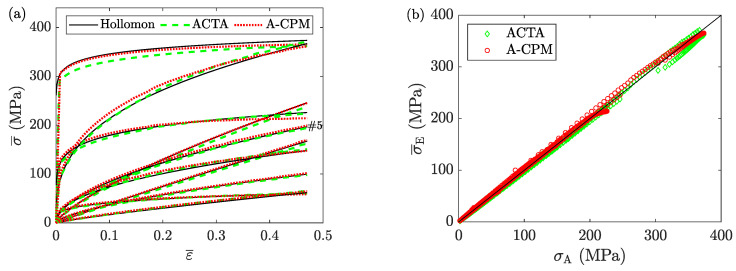
The stress analysis of the ten reference samples (see Table 1). (**a**) average flow curves obtained from A-CPM (Equation (Equation 4)) and ACTA (according to the algorithm developed for Case 2) in comparison with the assigned Hollomon models. (**b**) the correlation between the estimated average stresses (A-CPM and ACTA) and the assigned values. (Sample #5 is randomly selected for further detailed discussions in this section. For that, its flow curve is identified in the subset (**a**)).

**Figure 17 materials-16-00441-f017:**
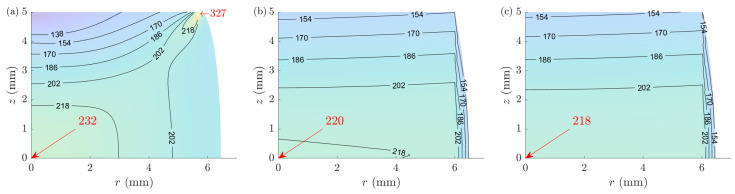
Comparison of the von Mises stress distributions in a randomly selected sample (#5) for (**a**) FEM and the proposed (**b**) ACTS and (**c**) ACTA models. (The presented solutions for ACTA are essentially the solutions of ACTS using the parameters identified from the ACTA analyses of the FE virtual experiments). The values are in the units of MPa.

**Figure 18 materials-16-00441-f018:**
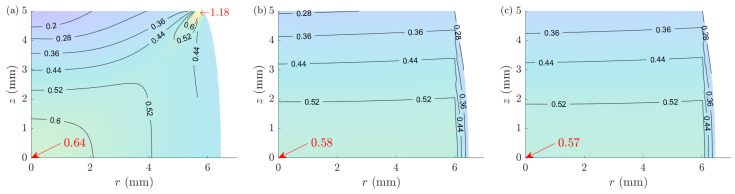
Comparison of the effective strain distributions in a randomly selected sample (#5) for (**a**) FEM, (**b**) ACTS, and (**c**) ACTA models.

**Figure 19 materials-16-00441-f019:**
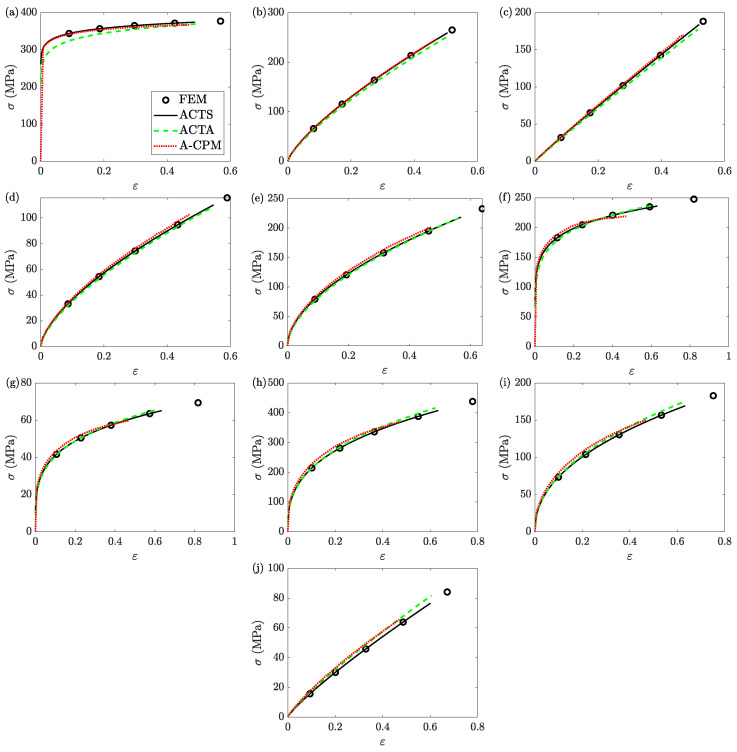
The stress-strain curves at the center of the samples (**a**) 1, (**b**) 2, (**c**) 3, (**d**) 4, (**e**) 5, (**f**) 6, (**g**) 7, (**h**) 8, (**i**) 9, and (**j**) 10. The data are obtained from different models of FEM, ACTS, and ACTA. The results of A-CPM, which are uniform throughout the sample, are added for comparison. Note that the ranges for both stress and strain vary for different samples, and $\max\left({\bar{\varepsilon }}_{A-\mathrm{CPM}}\right)=0.47$ for all samples.

**Table 1 materials-16-00441-t001:** The list of all samples investigated for the benchmark.

Sample	*m*	*k* (MPa)	*n*
1	0.05	388	0.05
2	0.15	433	0.75
3	0.25	343	0.95
4	0.35	163	0.65
5	0.45	298	0.55
6	0.55	253	0.15
7	0.65	73	0.25
8	0.75	478	0.35
9	0.85	208	0.45
10	0.95	118	0.85

**Table 2 materials-16-00441-t002:** The von Mises stress at the center of the sample in FEM, ACTS, ACTA, and A-CPM models, in the units of MPa. The numbers in the parentheses are the calculated percentage errors.

Sample	FEM	ACTS		ACTA		A-CPM	
1	377	374	(0.8)	370	(1.9)	374	(0.8)
2	265	260	(1.9)	250	(5.7)	246	(7.2)
3	188	184	(2.1)	178	(5.3)	167	(11.2)
4	116	110	(5.2)	108	(6.9)	100	(13.8)
5	232	220	(5.2)	218	(6.0)	197	(15.1)
6	248	237	(4.4)	239	(3.6)	226	(8.9)
7	69	66	(4.9)	66	(4.9)	60	(13.5)
8	438	411	(6.2)	419	(4.3)	367	(16.2)
9	183	172	(6.0)	177	(3.3)	148	(19.1)
10	84	78	(7.4)	83	(1.4)	62	(26.4)

**Table 3 materials-16-00441-t003:** The effective strain at the center of the FEM, ACTS, and ACTA models; the effective strain for the A-CPM model is uniform throughout each sample and the same for all samples, that is ${\bar{\varepsilon }}_{A-\mathrm{CPM}}=0.47$. The numbers in the parentheses are the calculated percentage errors.

Sample	FEM	ACTS		ACTA		A-CPM
1	0.57	0.49	(14.0)	0.50	(12.3)	(17.4)
2	0.52	0.51	(2.6)	0.50	(3.6)	(9.5)
3	0.53	0.52	(2.4)	0.52	(3.0)	(11.7)
4	0.59	0.55	(7.2)	0.54	(8.1)	(20.3)
5	0.64	0.58	(9.7)	0.57	(11.3)	(26.4)
6	0.82	0.65	(20.5)	0.64	(21.5)	(42.7)
7	0.82	0.65	(19.8)	0.64	(22.1)	(42.5)
8	0.78	0.66	(15.8)	0.64	(17.9)	(39.7)
9	0.75	0.65	(12.9)	0.65	(13.2)	(37.5)
10	0.67	0.61	(8.5)	0.62	(7.7)	(30.1)

**Table 4 materials-16-00441-t004:** The assigned ($m_{\mathrm{FEM}}$) and estimated friction factors of the ten samples used for the evaluation. The estimations for the Avitzur model are obtained from Equation (Equation 3) with its barreling parameter from Equation (Equation 30).

Sample	1	2	3	4	5	6	7	8	9	10
$m_{\mathrm{FEM}}$	0.05	0.15	0.25	0.35	0.45	0.55	0.65	0.75	0.85	0.95
$m_{\mathrm{ACTA}}$	0.07	0.13	0.22	0.31	0.41	0.54	0.58	0.68	0.85	1.00
$m_{\mathrm{Avitzur}}$	0.04	0.04	0.05	0.08	0.11	0.21	0.19	0.19	0.19	0.17

## Data Availability

The data presented in this study or their references are available within the article.

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
