# Peer review of "ACTAS: A New Framework for Mechanical and Frictional Characterization in Axisymmetric Compression Test"

_materials, 2023, doi:10.3390/ma16010441_

Round 1

Reviewer 1 Report

In this article, the phenomenon of barreling has been studied very deeply. The references are well used and the results are displayed with various graphs. Therefore, it will be possible to publish after minor revisions:

1-      It is recommended not to use abbreviations in subtitled. Therefore, for example “2.1. CPM” should be replaced by “2.1. Cylindrical Profile Model”.

2-      Due to the fact that in this new model (ACTAS), the equations are used in an incremental way and solved numerically, the results are not in the closed form equations, therefore, it is suggested that the results be presented as a series of calibration curves. In this way, it will be usable for others.

Reviewer 2 Report

The manuscript, entitled ‘ACTAS: A New Framework for Mechanical and Frictional Characterization in Axisymmetric Compression Test’ deals with the development of a new calculation method for cylindrical compression tests. Based on the detailed issues below, this Reviewer suggests minor revision of the manuscript.

- Please clarify the friction factor ‘m’ if it is Kudo or Coulomb when it is first appearing in the text.

- Are two strain rate components missing in equation (8) or is it on purpose that the  and  not included?

- In section 4.2 the shear stress is denoted with κ, why not with τ?

- In section 5. the Author state that the velocity is 2 mms-1, why did the Author choose such a big velocity? Also none of the material properties are dependent on the strain rate, so if my understanding is correct, this parameter is irrelevant.

- English is not the native language of this Reviewer, however a proof reading is suggested.

Reviewer 3 Report

The paper presents a numerical implementation that incorporate foldover to solve the problem of  Axisymmetric Compression Test. The validation of the algorithm is done by comparison with FE results of a commercial software.

The article is well presented but the conclusion should be, in my opinion,  restructured to underline the main findings of the paper and the novelty of the presented research.   
